# Classification Model for Diabetic Foot, Necrotizing Fasciitis, and Osteomyelitis

**DOI:** 10.3390/biology11091310

**Published:** 2022-09-03

**Authors:** Jiye Kim, Gilsung Yoo, Taesic Lee, Jeong Ho Kim, Dong Min Seo, Juwon Kim

**Affiliations:** 1Department of Plastic Surgery, Yonsei University Wonju College of Medicine, Wonju 26411, Korea; 2Department of Laboratory Medicine, Yonsei University Wonju College of Medicine, Wonju 26411, Korea; 3Division of Data Mining and Computational Biology, Institute of Global Health Care and Development, Wonju Severance Christian Hospital, Wonju 26411, Korea; 4Department of Family Medicine, Yonsei University Wonju College of Medicine, Wonju 26411, Korea; 5Center for Precision Medicine and Genomics, Wonju Severance Christian Hospital, Wonju 26411, Korea; 6Department of Medical Information, Yonsei University Wonju College of Medicine, Wonju 26411, Korea

**Keywords:** diabetic foot ulcer, necrotizing fasciitis, osteomyelitis, machine learning

## Abstract

**Simple Summary:**

Necrotizing fasciitis (NF) and osteomyelitis (OM) are severe complications in patients with diabetic foot ulcers (DFUs). Although NF and OM often cause results including limb amputation and death, definite diagnoses of these are challenging. To aid the prompt and proper diagnosis of NF and OM in patients with DFU, we developed and evaluated a novel prediction model based on machine learning technology. In summary, our prediction model appropriately discriminated the NF and OM from diabetic foot. Moreover, this prediction model has advantages in that it is based on the demographic data and routine laboratory results, which requires no additional examinations which are complicated or expensive.

**Abstract:**

Diabetic foot ulcers (DFUs) and their life-threatening complications, such as necrotizing fasciitis (NF) and osteomyelitis (OM), increase the healthcare cost, morbidity and mortality in patients with diabetes mellitus. While the early recognition of these complications could improve the clinical outcome of diabetic patients, it is not straightforward to achieve in the usual clinical settings. In this study, we proposed a classification model for diabetic foot, NF and OM. To select features for the classification model, multidisciplinary teams were organized and data were collected based on a literature search and automatic platform. A dataset of 1581 patients (728 diabetic foot, 76 NF, and 777 OM) was divided into training and validation datasets at a ratio of 7:3 to be analyzed. The final prediction models based on training dataset exhibited areas under the receiver operating curve (AUC) of the 0.80 and 0.73 for NF model and OM model, respectively, in validation sets. In conclusion, our classification models for NF and OM showed remarkable discriminatory power and easy applicability in patients with DFU.

## 1. Introduction

Diabetic foot ulcers (DFUs), one of the most common complications of diabetes mellitus (DM), lead to increased morbidity, mortality, and healthcare costs. Approximately 19–34% of patients with diabetes could be encountered with DFU in their lifetime, and these patients have a 2.5-fold increased risk of death at five years compared with those without foot ulcers [1]. Surprisingly, the total costs for diabetic foot care exceed those for many common cancers, including breast, colorectal, and lung cancers [2].

DFU is preceded by repetitive stress on the foot surface, with peripheral neuropathy and/or peripheral artery disease. The European Study Group on Diabetes and the Lower Extremity (EURODIALE) study reported that 58% of DFUs, and as high as 82% in hospitalized patients, are infected [3]. Once infected, the wound might heal poorly or deteriorate to the extent of requiring amputation [4]. Therefore, prompt recognition of infection and treatment with appropriate antibiotics can improve the outcomes of diabetic foot infections. The Infectious Disease Society of America (IDSA) recommends that wound infections should be diagnosed with at least two signs or symptoms of inflammation (erythema, warmth, tenderness, pain, and induration) or purulent secretion and be treated based on deep tissue culture results via biopsy or curettage rather than by swabs [5].

Necrotizing fasciitis (NF), recognized since the era of Hippocrates, is a rapidly progressive infection of the skin and soft tissues that leads to necrosis, systemic toxicity, and demise [6]. Diabetes is the most common comorbidity in patients with NF and has a prevalence ranging from 40% to 60% [7]. While NF can be diagnosed using a classic triad of symptoms (local pain, swelling, and erythema), its real-world application is challenging [8]. Further, a definitive diagnosis of NF requires laboratory and histopathologic findings, similar to that of other DFU infections. The various symptoms/statuses of patients with NF and the lack of early pathognomonic signs hinder identification of NF in the early phase.

Osteomyelitis (OM), arising from the spread of contiguous soft tissue infection and penetration through the cortical bone into the medullary cavity, occurs in 10–15% of moderate and in 50% of severe infections [9,10]. OM often requires surgical treatment with prolonged antibiotic therapy and poses an increased risk of limb amputation [11]. The presence of exposed bone, positive probe-to-bone test results, and imaging test results, including plain radiography and MRI, are helpful in diagnosing OM [12]. As DFU inherently has a chronic course and requires long-term therapy, a proper diagnostic imaging test could be delayed in diabetic foot OM [13]. If treatment is initiated when a definite diagnosis of OM is made based on clinical symptoms, the infection may have progressed too far to receive on-time treatment [14].

As prolonged and worsening DFU infections result in at least limb amputation, the importance of early recognition cannot be overemphasized [15]. However, the chronic course of DFU and the indistinct signs of its infectious complications hinder differential diagnosis. To solve this complicated issue, machine learning (ML) has the potential for developing a prediction model for DFU complications. Many efforts using ML have been made to improve the clinical course and treatment outcomes in patients with DFU. Schäfer et al. [16] used ML to predict the occurrence of DFU and amputation with risk factors, including the socioeconomic information of patients and their medical history. Goyal et al. [17] demonstrated that ML can properly recognize the presence of ischemia and infection in DFU images. Further, ML can be used to analyze foot thermogram images on smartphone applications for the early detection of DFU [18].

Unlike the common misconception that ML intuitively extracts plausible answers in response to specific inputs, ML develops and deploys algorithms for analyzing data and its characteristics and using statistical methods to determine the optimal outputs. In other words, ML algorithms represent a kind of mathematical model mapping a set of “features” (observed variables) onto a set of “labels” (outcome variables) [19]. Therefore, sophisticated designation of features or labels is essential for ML-based research. In this study, we initially investigated and reviewed the well-known risk factors for DFU infection by searching medical literature. Thereafter, we combined these results with actual electronic health records to embody an accurate prediction model for DFU infections. Finally, we developed and validated a classification model for DFU infection.

## 2. Materials and Methods

### 2.1. Participants

We retrospectively analyzed patients with a diabetic foot who visited Wonju Severance Christian Hospital (WSCH) between March 2012 and February 2021. Initially, we reviewed the following KCD-6 codes matched to diabetic foot: E10.7, E11.7, E12.7, E13.7, and E.14.7. Diagnosis of DFU infection was confirmed based on clinical findings and tissue culture results by experienced plastic surgeons, family physicians, and laboratory physicians based on the guidelines of the International Working Group on the Diabetic Foot (IWGDF) [20]. Among these patients, those with KCD-6 codes representing NF (M72.67) or OM (M86.96) were selected. The patients with NF were diagnosed with typical signs and symptoms such as tense edema outside the area of compromised skin, skin discoloration, blisters/bullae and necrosis, and crepitus and/or subcutaneous gas, combined with laboratory imaging, and microbiological findings by experienced plastic surgeons [21]. The diagnoses of OM also were established considering probe-to-bone test result, clinical, laboratory, and imaging findings by experienced plastic surgeons.

Initially 2349 datasets of patients with DFU were enrolled, and those with incomplete data for medical history or laboratory information were excluded. In total, 1581 patient datasets (728, 76, and 777 patients with diabetic foot, NF, and OM) were used to establish the prediction model and were randomly divided into training and validation datasets at a ratio of 7:3 (Figure 1A).

This study was approved by the Institutional Review Board (IRB) of WSCH (IRB No. CR322026). As the study was performed retrospectively with pre-existing medical records, the requirement for written consent from patients was waived, which was confirmed by the IRB. This study was conducted in accordance with the ethical principles of the Declaration of Helsinki. All enrolled individuals were processed anonymously and de-identified.

### 2.2. Selection of Predictors for Necrotizing Fasciitis and Osteomyelitis

To establish a classification model for DFU infections, we used the Bayesian approach, which provides two major advantages. Bayesian manners yield transparency by offering the complete probability distributions for the estimated model parameters, statistical metrics, and predictions. Further, Bayesian models can easily incorporate previously available scientific information into new data [15]. Subsequently, the selection of plausible predictors (also termed features) was considered a crucial task based on previous studies. In fact, several studies used a literature-search approaches to identify disease-related features or predictors [22,23,24]. Motivated by these studies, two experts in plastic surgery and family medicine reviewed the literature and yielded approximately 30 variables known to be related to NF and/or OM (Appendix A). Then, the database administrator extracted the automatic platform-based features from the electronic health records (EHR) at the WSCH. Finally, the candidate features obtained from the literature search and EHR were evaluated using a statistical model (Figure 1B).

Among numerous statistical methods for selecting features, the stepwise feature selection could be divided into forward selection or backward elimination [25]. We implemented the modified version of backward elimination for feature selection. Although typical backward stepwise elimination sequentially removes a feature with the most insignificant result one by one, our modified backward elimination method subtracted all features exhibiting insignificant finding (*p* < 0.1 in multivariate LR) at once. The modified backward elimination method used in our study has been attempted in previous studies [22,23,26].

### 2.3. Establishment of a Prediction Model

Numerous statistical approaches have been used for feature identification. Lee and Lee [27] integrated multiple statistical methods, such as the *t*-test and correlation method (i.e., the biweight midcorrelation method), to identify features. Moon et al. [26] initially screened risk factors based on expert knowledge and finally determined predictors using multiple steps of statistical methods, including logistic regression (LR). LR is a frequently used approach for predicting DFU infections [28,29]. Furthermore, efforts have been made to establish a prediction model for NF and OM using LR [30,31]. Likewise, we could establish the classification model for DFU infections using LR based on the ML technique. This LR model consists of linear units and non-linear unit referred to as the ‘sigmoid function’.

### 2.4. Statistics

Differences in variables were analyzed based on DFU infection status using Student’s *t*-test and Chi-square test for continuous and categorical variables, respectively. The DFU infection prediction model was evaluated in terms of performance using the receiver-operating characteristic (ROC) curve and the area under the curve (AUC), which is a combination of sensitivity and specificity. Statistical analysis was performed using R language (R package ver. 4.1.2, R Foundation for Statistical Computing, Vienna, Austria). The *p*-values < 0.05 were considered statistically significant.

## 3. Results

The general characteristics of the datasets are presented in Table 1. NF showed the following differential characteristics compared to diabetic foot: low ratio of females; increased levels of C-reactive protein (CRP), white blood cells (WBC), mean platelet volume (MPV), delta neutrophil index (DNI), myeloperoxidase index (MPXI), and neutrophil-lymphocyte ratio (NLR), and decreased levels of creatinine (Cr), total protein (TP), Ca, K, HbA1c, and platelets (PLT). In addition, the following OM-related characteristics were observed: higher TP, Ca, Na, Cl, red blood cells (RBC), hemoglobin (Hb), and hematocrit (Hct), and lower age, female ratio, CRP, blood urea nitrogen (BUN), Cr, K, HbA1c, erythrocyte sedimentation rate (ESR), WBC, MPV, DNI, and NLR.

Among the 28 risk factors (referred to as literature search-based features) related to NF or OM obtained from the literature-based search, 22 predictors were present in the automatic platform (Table 1). Therefore, we processed these 22 variables using univariate and multivariate LR analyses (also termed stepwise LR) to identify the features of NF or OM prediction models (Table 2 and Table 3).

In the univariate LR for NF status, 11 of the 22 predictors were preliminarily selected (Table 2) and processed into a multivariate model. Notably, when identifying predictors for the classification model, *p* < 0.1 was implemented in univariate or multivariate LR models. Seven predictors were identified as final input variables for the NF prediction model. Female sex, CRP, DNI, and NLR were positively associated with NF status (vs. diabetic foot), and the remaining three variables were negatively related to NF (Table 2).

In the OM prediction model, 12 predictors were selected as final input variables. Among these 12 variables, younger age, female sex, TP, Cl, Hct, and PLT were positively correlated with OM status (Table 3).

The odds ratios for each feature in NF and OM (Table 2 and Table 3) were log-transformed, followed by the establishment of the DFU infection prediction model described in Table 4. From these weights (coefficients), index values for the probability of DFU infections, ranging from 0 to 1, were calculated and applied to the validation dataset. As a result, areas under the receiver operating curve (AUC) of 0.80 and 0.73 were obtained from the NF and OM prediction models for the validation sets, respectively (Figure 2). Based on the maximum value of the F-measure, the optimal cut-offs for NF and OM were determined as 0.13 and 0.3, respectively.

## 4. Discussion

We identified predictors for DFU infections and proposed a novel classification model based on a literature search and automatic platform data using the LR method. Although Wong et al. [32] proposed the Laboratory Risk Indicator for Necrotizing Fasciitis (LRINEC) scoring system for differentiating NF from other infections in 2004, this system could not maintain outstanding performance in various studies [33,34]. Considering the fulminant and dismal course of NF, a reliable and robust method is required for early differential diagnosis of patients with DFU. Further, OM, which is more prevalent and challenging than NF for clinicians managing patients with diabetic foot, should not be overlooked. Our classification model could discriminate NF and OM from other DFU infections, despite being made of easily and cheaply obtainable parameters including demographic data and routine laboratory results, rather than state-of-the-art or costly markers.

The IWGDF guidelines recommend the use of inflammatory biomarkers such as WBC, ESR, CRP, and procalcitonin to establish a diagnosis of diabetic foot infection [20]. Among the inflammatory markers included in our study, CRP, DNI, and NLR showed a significant positive correlation with NF in the multivariate analysis, whereas ESR and MPXI did not. In general, CRP is known to have a higher diagnostic accuracy for infection than that of WBC or ESR [20]. Moreover, a recent study demonstrated that the DNI and NLR are robust predictors of equivocal septic conditions using clustering analysis [24]. In contrast, the OM group exhibited significantly negative associations only with ESR and NLR. Several studies have revealed that ESR is the most useful marker for bone infection, except that it shows an opposite trend [35,36]. Similarly, Serban et al. [37] found that elevated NLR is correlated with OM in DFU infections. The negative correlation between ESR, NLR, and OM in our study, which is inconsistent with previous results, should thus be evaluated further.

In the EURODIALE study, male sex was found to be an independent predictor of non-healing DFU [38]. Similar findings have been reported in other populations [39,40]. Therefore, this could be partially explained by the hypothesis that prolonged non-healing DFU, which is more common in male patients, is likely to be the point of pathogen entry and consequently cause infections [28,41]. Likewise, male predominance was observed in all three groups in our study, while both the NF and OM groups exhibited relative female predilections compared to those of the diabetic foot group. There are conflicting data regarding whether sex affects the development of DFU infection. An Australian cohort study proposed that female sex was a risk factor for DFU infection, consistent with the results of our study [42]. Nevertheless, a recent meta-analysis argued that sex does not affect the development of osteomyelitis in patients with DFU [43].

Further, age showed a significantly negative association in the OM group, consistent with a previous study by Lavery et al. [44] showing that older patients (age ≥ 70 years) had reduced osteomyelitis (relative risk = 0.46). Similarly, the aforementioned Australian study revealed that younger age is a risk factor for developing DFU infections [42]. Although OM has distinct differences in the major routes of infections and causative organisms according to patient age, our dataset has limitations in considering these factors [45]. Therefore, future studies that consider these factors are needed to verify the negative correlation between OM and age observed in our study.

Researchers of ML-based studies should understand the trade-off relationship between explainability and training accuracy among many ML models and select the relevant model for the intended goals [46]. Owing to insufficient research on ML-based classification models for DFU infections, we used the LR model, which is highly interpretable despite its lower accuracy, in our study. Therefore, several discordant findings with previous studies require verification through further research using more accurate ML models such as support vector machines, random forests, and deep neural networks [47].

For example, although increased serum creatinine was a positive predictor of the LRINEC score, it was a negative predictor for both NF and OM in our study population [32]. This result is contrary to Game’s theory that inflammation associated with DFU induces a decline in renal function [48]. We postulated that the discrepancy with Wong’s study [32] could partly have resulted from difference in the control group (e.g., cellulitis or abscesses vs. DFU). Furthermore, our study revealed that HbA1c levels are negatively associated with both NF and OM. Increased levels of HbA1c, a surrogate marker for poor glycemic control, have usually been employed to predict adverse outcomes such as lower extremity amputation or mortality in diabetic infections [49,50]. The cross-sectional nature of our study could be implicated in this questionable result. Given the fact that all three groups in our study showed the fairly higher level of HbA1c than 7%, the optimal glycemic target recommended by American Diabetes Association [51], the association of HbA1c with NF and/or OM could be different in prospectively well-controlled glycemic cohort. Lastly, lower MPV was associated with OM in our study, whereas MPV is often increased in many inflammatory conditions, including cardiovascular diseases, cerebral stroke, respiratory diseases, chronic renal failure, intestinal diseases, rheumatoid diseases, diabetes, and various cancers [52].

The present study has some limitations. First, the patients enrolled in this study were from a single tertiary hospital in South Korea, which makes it difficult to apply this classification model to every region or ethnicity. Further study based on multi-center or multi-ancestry cohort is required to obtain more generalized and improved prediction models for NF and OM. Second, we designed the study using only a cross-sectional dataset and did not include longitudinal trends or changes in predictors. Therefore, it is desirable that the diagnostic performance of the proposed classification model should be evaluated in a prospective cohort study to identify the causality of the predictors in our models and validate whether the cross-sectional data-derived model could truly predict new-onset cases [23]. Third, a relatively small number of patients with NF were included compared to those with diabetic foot and OM. However, this rarity is natural given the low incidence of NF (0.03–2.17 per 100,000 population) in Korea [53]. Moreover, the number of patients with NF in our cohort is large enough to establish a robust prediction model, compared with previous NF studies [34]. Fourth, the microbiological results and clinical outcomes such as low extremity amputation rates and mortality were not analyzed owing to the incompleteness and heterogeneity of the data. In our opinion, the proposed prediction model in this study could be combined with the microbiological and clinical outcome profiles in well-designed prospective cohort in the future. By extension, further study based on a clustering analysis method comparing the combined model with pre-existing etiology-based NF classification (type I~IV) is feasible and promising [24,54]. Lastly, we did not consider other comorbidities affecting the infection status. For instance, Furuse et al. [55] suggested that chronic hepatitis could be a risk factor for NF and should be included in the LRINEC scoring system.

## 5. Conclusions

In conclusion, we propose a novel classification model for DFU infections, composed of several common indices in routine clinical settings based on multidisciplinary approaches, including various departments of clinicians and experts on automatic platforms. This classification model might pave the way for the early discrimination of severe infectious complications and improve the clinical outcome and prognosis in patients with DFU.

## Figures and Tables

**Figure 1 biology-11-01310-f001:**
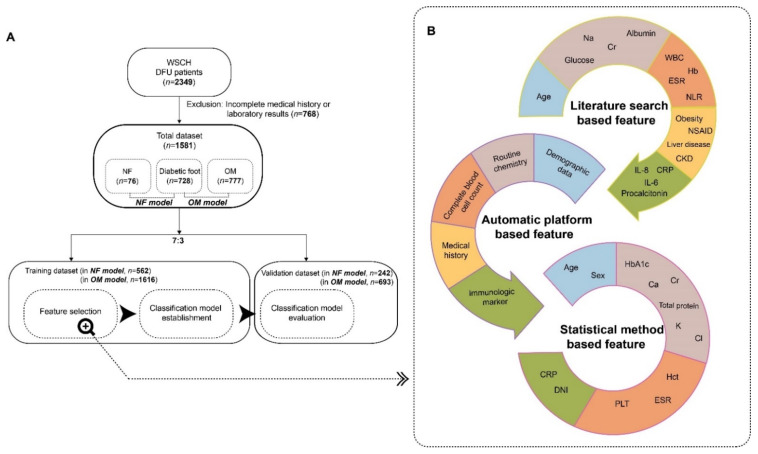
The development and evaluation for a classification model for DFU infections. (**A**) The workflow of entire study including patient data collection, feature selection, classification model establishment and validation. (**B**) The detailed feature selection process for establishment of prediction model. The features for prediction model were screened and selected based on integrative tasks between literature search and automatic platform. The final predictors were selected based on the statistical method.

**Figure 2 biology-11-01310-f002:**
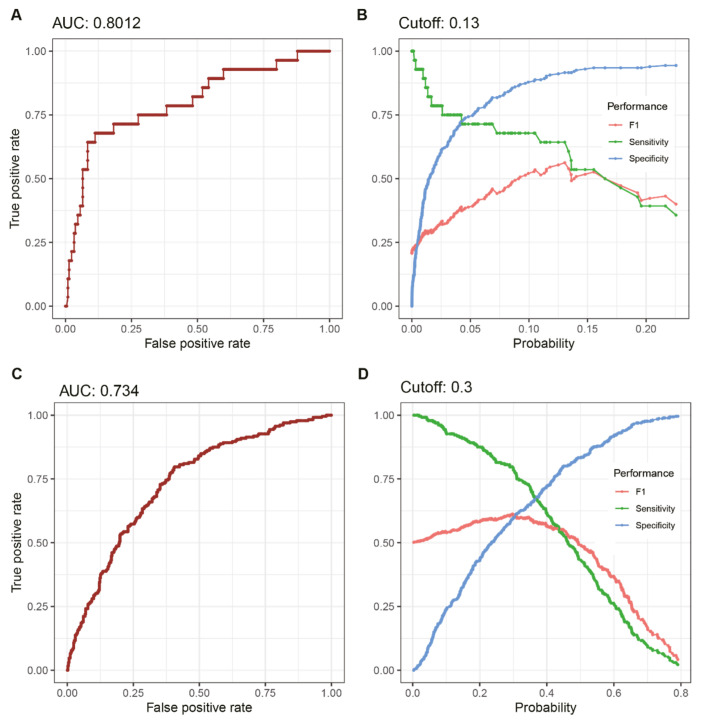
Classification performance of the prediction model for necrotizing fasciitis and osteomyelitis. Among a total of 1581 diabetic foot patients, 70% were randomly assigned as the “training dataset” for feature selection and establishment of the classification model. (**A**) The performance of the LR model was evaluated using the other patients as the “validation dataset.” The area under the ROC curve of the LR model for necrotizing fasciitis was 0.8012. (**B**) The optimal cut off (0.13) for necrotizing fasciitis was determined by the maximum value in the F-score curve based on sensitivity and specificity. (**C**) The area under the ROC curve of LR model for osteomyelitis was 0.734. (**D**) The optimal cut off for osteomyelitis was 0.3. ROC, receiver operating characteristic.

**Table 1 biology-11-01310-t001:** General characteristics of diabetic foot ulcer infections.

	Diabetic Foot (Ref)	Necrotizing Fasciitis *(p*-Value, vs. Ref)	Osteomyelitis *(p*-Value, vs. Ref)
N	728	76	777
Age, years	70.2 ± 0.47	69.8 ± 1.55 (0.788)	66.9 ± 0.49 (<0.001)
Sex (Male), n	539 (74.0%)	42 (55.3%, 0.001)	442 (56.9%, <0.001)
CRP, mg/dL	4.9 ± 0.26	12.9 ± 1.32 (<0.001)	3.5 ± 0.41 (<0.001)
BUN, mg/dL	26.6 ± 0.69	24.7 ± 1.91 (0.349)	17.9 ± 0.6 (<0.001)
Creatinine, mg/dL	2.4 ± 0.1	1.3 ± 0.12 (<0.001)	1.2 ± 0.04 (<0.001)
Total protein, g/dL	6.6 ± 0.03	5.9 ± 0.13 (<0.001)	6.8 ± 0.04 (<0.001)
Ca, mg/dL	8.8 ± 0.03	8.3 ± 0.11 (<0.001)	9.1 ± 0.03 (<0.001)
Na, mmol/L	137.3 ± 0.18	136.9 ± 0.74 (0.664)	138.7 ± 0.23 (<0.001)
K, mmol/L	4.5 ± 0.03	4.1 ± 0.08 (<0.001)	4.3 ± 0.02 (<0.001)
Cl, mmol/L	101 ± 0.21	101.5 ± 0.77 (0.603)	102.8 ± 0.24 (<0.001)
HbA1c, %	8.3 ± 0.08	7.8 ± 0.22 (0.031)	7.9 ± 0.07 (<0.001)
ESR, mm/h	50 ± 1.19	53.7 ± 3.6 (0.339)	39.5 ± 1.13 (<0.001)
WBC, ×10^9^/L	10.1 ± 0.22	16 ± 1.29 (<0.001)	9.1 ± 0.4 (<0.001)
RBC, ×10^12^/L	3.8 ± 0.03	3.8 ± 0.09 (0.654)	4.2 ± 0.03 (<0.001)
Hemoglobin, g/dL	11.6 ± 0.08	11.6 ± 0.27 (0.859)	12.6 ± 0.08 (<0.001)
Hematocrit, %	34.8 ± 0.22	34.8 ± 0.81 (0.962)	37.6 ± 0.25 (<0.001)
Platelet, ×10^9^/L	280.2 ± 4.42	242.1 ± 15.66 (0.021)	288.7 ± 4.9 (0.158)
MPV, fL	8 ± 0.04	8.4 ± 0.16 (0.005)	7.7 ± 0.05 (<0.001)
MPC, g/dL	26.5 ± 0.07	26.5 ± 0.17 (0.975)	26.6 ± 0.05 (0.354)
DNI, %	0.9 ± 0.11	9.1 ± 1.62 (<0.001)	0.7 ± 0.51 (0.029)
MPXI	0.1 ± 0.17	1.8 ± 0.53 (0.003)	0.4 ± 0.17 (0.172)
NLR, %	7.8 ± 0.46	24.6 ± 2.7 (<0.001)	5 ± 0.84 (<0.001)

Continuous variables are presented as mean ± standard deviation, and categorical variables are presented as number and ratio. DM, diabetes mellitus; CRP, C-reactive protein; BUN, blood urea nitrogen; ESR, erythrocyte sedimentation rate; WBC, white blood cell; RBC, red blood cell; MPV, mean platelet volume; MPC, mean platelet component; DNI, delta neutrophil index; MPXI, myeloperoxidase index; NLR, neutrophil–lymphocyte ratio.

**Table 2 biology-11-01310-t002:** Feature selection of prediction model for necrotizing fasciitis using backward stepwise logistic regression.

	Univariate	Multivariate (Model 1)	Multivariate (Model 2)
	OR (95% CI)	OR (95% CI)	OR (95% CI)
Age	1.003 (0.98–1.026)	NS	NS
Sex (Female)	3.42 (1.875–6.24)	5.161 (2.183–12.203)	5.394 (2.3–12.65)
CRP, mg/dL	1.103 (1.07–1.137)	1.07 (1.017–1.125)	1.07 (1.021–1.121)
BUN, mg/dL	0.993 (0.975–1.011)	NS	NS
Creatinine, mg/dL	0.716 (0.551–0.932)	0.486 (0.305–0.774)	0.482 (0.305–0.763)
Total protein, g/dL	0.399 (0.285–0.558)	0.864 (0.46–1.623)	NS
Ca, mg/dL	0.397 (0.276–0.57)	0.566 (0.289–1.107)	0.49 (0.299–0.803)
Na, mmol/L	1 (0.939–1.064)	NS	NS
K, mmol/L	0.36 (0.218–0.594)	0.708 (0.354–1.416)	NS
Cl, mmol/L	1.037 (0.983–1.094)	NS	NS
HbA1c, %	0.791 (0.666–0.939)	0.79 (0.641–0.973)	0.767 (0.626–0.939)
ESR, mm/h	1.006 (0.997–1.015)	NS	NS
WBC, ×10^9^/L	1.078 (1.037–1.121)	0.952 (0.882–1.027)	NS
RBC, ×10^12^/L	0.924 (0.607–1.407)	NS	NS
Hemoglobin, g/dL	0.959 (0.834–1.103)	NS	NS
Hematocrit, %	0.996 (0.949–1.046)	NS	NS
Platelet, ×10^9^/L	0.998 (0.996–1.001)	NS	NS
MPV, fL	1.243 (0.956–1.616)	NS	NS
MPC, g/dL	0.968 (0.833–1.125)	NS	NS
DNI, %	1.258 (1.157–1.369)	1.137 (1.063–1.216)	1.14 (1.065–1.22)
MPXI	1.097 (1.025–1.174)	0.991 (0.907–1.083)	NS
NLR, %	1.072 (1.05–1.093)	1.072 (1.031–1.114)	1.055 (1.025–1.085)

OR, odds ratio; CI, confidence interval; CRP, C-reactive protein; BUN, blood urea nitrogen; ESR, erythrocyte sedimentation rate; WBC, white blood cell; RBC, red blood cell; MPV, mean platelet volume; MPC, mean platelet component; DNI, delta neutrophil index; MPXI, myeloperoxidase index; NLR, neutrophil–lymphocyte ratio; NS, not selected.

**Table 3 biology-11-01310-t003:** Feature selection of prediction model for osteomyelitis using backward stepwise logistic regression.

	Univariate	Multivariate (Model 1)	Multivariate (Model 2)
	OR (95% CI)	OR (95% CI)	OR (95% CI)
Age	0.984 (0.978–0.991)	0.985 (0.977–0.993)	0.984 (0.977–0.992)
Sex (Female)	2.2 (1.769–2.735)	2.547 (1.961–3.308)	2.597 (2.011–3.353)
CRP, mg/dL	0.954 (0.937–0.971)	1.011 (0.988–1.036)	NS
BUN, mg/dL	0.953 (0.944–0.963)	0.992 (0.978–1.005)	NS
Creatinine, mg/dL	0.679 (0.615–0.75)	0.87 (0.779–0.97)	0.839 (0.77–0.915)
Total protein, g/dL	1.538 (1.355–1.746)	1.232 (1.025–1.48)	1.282 (1.102–1.491)
Ca, mg/dL	1.791 (1.539–2.084)	1.134 (0.907–1.418)	NS
Na, mmol/L	1.089 (1.062–1.117)	1.001 (0.959–1.046)	NS
K, mmol/L	0.66 (0.554–0.785)	0.618 (0.497–0.769)	0.611 (0.496–0.752)
Cl, mmol/L	1.073 (1.051–1.095)	1.05 (1.012–1.09)	1.047 (1.021–1.073)
HbA1c, %	0.934 (0.886–0.984)	0.881 (0.826–0.941)	0.88 (0.825–0.938)
ESR, mm/h	0.99 (0.986–0.993)	0.995 (0.99–1)	0.996 (0.991–1.001)
WBC, ×10^9^/L	0.961 (0.942–0.981)	1.019 (0.983–1.056)	NS
RBC, ×10^12^/L	1.925 (1.648–2.248)	0.805 (0.496–1.307)	NS
Hemoglobin, g/dL	1.232 (1.171–1.297)	0.937 (0.7–1.255)	NS
Hematocrit, %	1.076 (1.057–1.096)	1.11 (0.995–1.238)	1.067 (1.042–1.092)
Platelet, ×10^9^/L	1.001 (1–1.002)	1.002 (1–1.003)	1.002 (1.001–1.003)
MPV, fL	0.704 (0.627–0.79)	0.87 (0.756–1.001)	0.87 (0.761–0.996)
MPC, g/dL	1.029 (0.978–1.084)	NS	NS
DNI, %	0.91 (0.86–0.964)	0.981 (0.935–1.029)	NS
MPXI	1.008 (0.984–1.031)	NS	NS
NLR, %	0.944 (0.927–0.962)	0.972 (0.948–0.996)	0.977 (0.96–0.994)

OR, odds ratio; CI, confidence interval; CRP, C-reactive protein; BUN, blood urea nitrogen; ESR, erythrocyte sedimentation rate; WBC, white blood cell; RBC, red blood cell; MPV, mean platelet volume; MPC, mean platelet component; DNI, delta neutrophil index; MPXI, myeloperoxidase index; NLR, neutrophil–lymphocyte ratio.

**Table 4 biology-11-01310-t004:** Final prediction model for DFU infections.

Necrotizing Fasciitis		Osteomyelitis	
Predictors	Beta-Coefficient	Predictors	Beta-Coefficient
Constants	2.733	Constants	−5.218
Sex (Female)	1.685	Sex (Female)	0.954
CRP, mg/dL	0.068	Age	−0.016
Creatinine, mg/dL	−0.73	Creatinine, mg/dL	−0.175
Ca, mg/dL	−0.713	Total protein, g/dL	0.248
HbA1c, %	−0.266	HbA1c, %	−0.128
DNI, %	0.131	K, mmol/L	−0.493
NLR, %	0.053	Cl, mmol/L	0.046
		Hematocrit, %	0.065
		MPV, fL	−0.139
		Platelet, ×10^9^/L	0.002
		ESR, mm/h	−0.004
		NLR, %	−0.023

CRP, C-reactive protein; DNI, delta neutrophil index; NLR, neutrophil–lymphocyte ratio; MPV, mean platelet volume; ESR, erythrocyte sedimentation rate.

## Data Availability

Not applicable.

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
