# Peer review of "Classification Model for Diabetic Foot, Necrotizing Fasciitis, and Osteomyelitis"

_biology, 2022, doi:10.3390/biology11091310_

Round 1

Reviewer 1 Report

In this study, the authors proposed a classification model for diabetic foot, NF and OM in a retrospective observational study. Despite the limitations related to a retrospective observational study, they used proper methodology for and a large case series that answers the study question. The section on statistical analysis does not indicate the correct methodology used for step-wise regression. After clarification I believe the study can be approved.

Reviewer 2 Report

- Results need to be tested in a prospective multicenter trial to be approved for clinical practice as mentioned by the authors. 

- The contradiction with available scoring systems like the (LRINEC) scoring system, the significance of creatinine and HbA1c needs more explanation in the discussion section to clarify these paradoxical results.

- Need to explain the significance of the results for the necrotizing fasciitis group of patients considering the small number (76) of this group of patients.

- The necrotizing fasciitis selection criteria need to be mentioned properly.

-  No mention of any microbiological culture for encountered bacteria. 

- Amputation &/or death are significant outcome measures for diabetic foot infection, however, these results are not mentioned in the data sets.
